# Evaluation of the basic assay performance of the GeneSoc® rapid PCR testing system for detection of severe acute respiratory syndrome coronavirus 2

Ryosuke Watanabe[1], Satomi Asai [2,3]*, Hidehumi Kakizoe[2], Hirofumi Saeki[2], Atsuko Masukawa[1], Miki Miyazawa[3], Kazumi Ohtagawa[1], Mend-Amar Ravzanaaadii[2], Mika Doi[2], Haruyo Atsumi[2], Kazuo Umezawa[4], Hayato Miyachi[2,3]

1 Clinical Laboratory Center, Tokai University Hospital, Kanagawa, Japan, 2 Department of Laboratory Medicine, Tokai University School of Medicine, Isehara Kanagawa, Japan, 3 Division of Infection Control, Tokai University Hospital, Kanagawa, Japan, 4 Department of Emergency and Critical Care, Tokai University School of Medicine, Isehara, Kanagawa, Japan

* sa@is.icc.u-tokai.ac.jp

**Data Availability Statement:** All relevant data are within the paper.

## Abstract

In the ongoing coronavirus disease 2019 (COVID-19) pandemic, PCR has been widely used for screening patients displaying relevant symptoms. The rapid detection of severe acute respiratory syndrome coronavirus 2 (SARS-CoV-2) enables prompt diagnosis and the implementation of proper precautionary and isolation measures for the patient. In the present study, we aimed to evaluate the basic assay performance of an innovative PCR system, GeneSoC® (Kyorin Pharmaceutical Co. Ltd., Tokyo, Japan). A total of 1,445 clinical samples were submitted to the clinical laboratory, including confirmed or suspected cases of COVID-19, from February 13 to August 31. Specimen types included nasopharyngeal swabs. The sampling was performed several times for each patient every 2–7 days. Using this system, sequences specific for SARS-CoV-2 RNA could be detected in a sample within 10–15 min using the microfluidic thermal cycling technology. Analytical sensitivity studies showed that GeneSoC® could detect the target sequence of the viral envelope and RNA-dependent RNA-polymerase (RdRp) genes at 5 and 10 copies/μL, respectively. The precision of the GeneSoC® measurements using clinical isolates of the virus at a concentration of $10^3$ copies/μL was favorable for both the genes; within-run repeatability and between-run reproducibility coefficient of variation values were less than 3% and 2%, respectively; and the reproducibility of inter-detection units was less than 5%. Method comparison by LightCycler® 480 showed the positive and negative agreement to be 100% [(174/174) and (1271/1271), respectively]. GeneSoC® proved to be a rapid and reliable detection system for the prompt diagnosis of symptomatic COVID-19 patients and could help reduce the spread of infections and facilitate more rapid treatment of infected patients.

**Funding:** SA Study No. JP19he2202007 Japan Agency for Medical Research and Development (AMED) https://www.amed.go.jp/ The funders had no role in the study design, data collection and analysis, writing of the manuscript, or decision to publish this manuscript.

**Competing interests:** The authors have declared that no competing interests exist.

## Introduction

The novel coronavirus disease was initially reported as pneumonia of unknown origin occurring in the city of Wuhan in China's Hubei Province on December 30, 2019 [1]. After it was identified as a new type of coronavirus on January 7, 2020, it began to spread around the world, starting in Asia [2]. On February 11, 2020, the World Health Organization (WHO; https://www.who.int/) named this disease coronavirus disease 2019 (COVID-19) and gave the causal virus the official name of severe acute respiratory syndrome coronavirus 2 (SARS-CoV-2; https://talk.ictvonline.org/).

The disease frequently presents with symptoms, such as fever, respiratory symptoms, headaches, and malaise, with digestive symptoms appearing in less than 10% of cases. Approximately 80% of patients recover within one week and have only mild symptoms. For the remaining 20%, pneumonia symptoms worsen in 7–10 days, requiring hospitalization. In approximately 2%–3% of patients, symptom severity increases after 10 days, eventually causing the disease to become fatal [3, 4].

Currently, most qualitative tests for the detection of SARS-CoV-2 in Japan are carried out in accordance with the protocol recommended by the National Institute of Infectious Diseases (https://www.niid.go.jp/niid/ja/) using an equipment, such as the LightCycler® 480 system (Roche Diagnostics, Basel, Switzerland). However, the amplification and detection take at least 3 h, and the nucleic acid extraction process requires 30–60 min, resulting in a long total testing time of 4–5 h. In addition, the measurement operations are so complex that dedicated personnel with relevant competence are necessary. Because the assay is a batch process, individual real-time measurements at clinical testing sites are difficult. Recently, the innovative PCR system GeneSoC® (Kyorin Pharmaceutical Co. Ltd., Tokyo, Japan) that detects specific SARS-COV-2 RNA sequences using a microfluidic thermal cycling technology, has been developed. This system is capable of detecting a sequence specific to microbial pathogens in a sample within 10–15 min and can be operated on-site in a clinical setting [5, 6].

Here, we report the results of our study conducted to evaluate the basic assay performance of this innovative GeneSoC® system for the detection of SARS-CoV-2. The performance of the GeneSoC® system will exceed that of current test methods, both in terms of rapidity and reliability.

## Materials and methods

### Materials

A total of 1,445 clinical samples were submitted to the clinical laboratory upon testing requests from in-hospital patients and the outpatient department, including confirmed or suspected cases of COVID-19, from February 13 to August 31, 2020. Specimen types included nasopharyngeal swabs. The sampling was performed several times in each patient every 2–7 days. The patients provided verbal as well as written informed consent to participate in this study. This study was approved by the review board of Tokai University (19R-321).

### Specimen processing

Swab specimens were soaked and dissolved in 2.0 mL of PBS (-) immediately after collection. A total of 140 μL of this solution was applied to a QIAamp Viral RNA Mini Kit (Qiagen, Tokyo, Japan), and RNA was extracted following the manufacturer's instructions.

### PCR with GeneSoC®

GeneSoC® uses heaters to control the temperature for reverse transcription, denaturation, annealing, and extension processes, with thermal cycling performed by sending the PCR

solution to and from each heater. Accordingly, this microfluidic thermal cycling through the roundtrip transport of preheated denaturation, annealing, and extension processes drastically reduces the time required for temperature changes in the sample solution. The GeneSoC® device is composed of a main unit and a detection unit. The system is expandable to run up to four detection units on a single main unit. All components of the system are integrally controlled [5, 6]. The reaction kit and polymerase used for this study were the One Step Prime Script RT-PCR kit (Takara Bio Inc., Shiga, Japan) and Speed STAR HS DNA polymerase (Takara Bio Inc.), as described in Table 1A and 1B. Reaction reagent (15 μL) was mixed with template RNA (5 μL), and the solution was dispensed onto a reaction panel chip customized for GeneSoC®. Primers and probes were targeted to E genes (hereinafter, E primer) and RdRp genes (hereinafter, RdRp primer). The primer and probe arrangement (5′ to 3′) (Table 2) [7] and PCR conditions (Table 3) were as shown.

## PCR with LightCycler® 480

The PCR method using the LightCycler® 480 system is the major testing method used in Japan, and for this study, it was performed in accordance with the Novel Coronavirus Pathogen Detection Manual issued by the National Institute of Infectious Diseases (https://www.niid.go.jp/niid/ja/). The kit used was the QuantiTect Probe RT-PCR kit (Qiagen). Reaction reagent (15 μL) was mixed with template RNA (5 μL), and the solution was dispensed on the LightCycler® 96 well plate. Primers and probes were targeted to N2 genes. Although there are two probes for the N and N2 regions, the more sensitive probe N2 was chosen for comparison. The primer and probe arrangement (5′ to 3′) (Table 4) [8] and PCR conditions (Table 5) were as shown.

## Analytical sensitivity

The AcroMetrix™ Coronavirus 2019 (COVID-19) RNA Control (Thermo Fisher Scientific Waltham, MA, USA) was used for the evaluation of analytical sensitivity, and a serial dilution

**Table 1. A. Components of high-speed RT-PCR mixtures (for the E gene).** B. Components of high-speed RT-PCR mixtures (for the RdRp Gene).

| A. | | |
|---|---|---|
| **Reagents** | | **Final concentration** |
| 2X OneStep RT-PCR Buffer III[a] | | 1X |
| 50X ROX solution[a] | | 0.2X |
| PrimeScript RT enzyme Mix II[a] | | 2 U/μL |
| SpeedSTAR™ HS DNA Polymerase | | 0.25 U/μL |
| E primer solution | Forward | 2.0 μM |
| | Reverse | 2.0 μM |
| E probe solution | | 0.2 μM |
| **B.** | | |
| **Reagents** | | **Final concentration** |
| 2X OneStep RT-PCR Buffer III[a] | | 1X |
| 50X ROX solution[a] | | 0.2X |
| PrimeScript RT enzyme Mix II[a] | | 2 U/μL |
| SpeedSTAR™ HS DNA Polymerase | | 0.25 U/μL |
| RdRp primer solution | Forward | 2.4 μM |
| | Reverse | 3.2 μM |
| RdRp probe solution | | 0.4 μM |

[a]Each reagent was included in the One Step PrimeScript™ RT-PCR Kit (Takara Bio Inc., Shiga, Japan).

**Table 2. GeneSoC® primer and probe arrangement.**

| Primer | Primer and Probe Arrangement (5′ to 3′) |
|---|---|
| E gene Forward (2.0 μM) | ACAGGTACGTTAATAGTTAATAGCGT |
| E gene Reverse (2.0 μM) | ATATTGCAGCAGTACGCACACA |
| Probe (0.2 μM) | Cy5-ACACTAGCCATCCTTACTGCGCTTCG-BHQ3 |
| RdRp gene Forward (2.4 μM) | GTGARATGGTCATGTGTGGCGG |
| RdRp gene Reverse (3.2 μM) | CARATGTTAAASACACTATTAGCATA |
| Probe (0.4 μM) | Cy5-CAGGTGGAACCTCATCAGGAGATGC-BHQ3 |

method was carried out using RNase-free water ($5 \times 10^1$, $4 \times 10^1$, $3 \times 10^1$, $2 \times 10^1$, $1 \times 10^1$, $5 \times 10^0$, and $2 \times 10^0$ copies/μL). The analytical sensitivity of GeneSoC® (E primer and RdRp primer) was compared with that of LightCycler® 480 (N2 primer). The required volume of control RNA was included in each assay (in units of 5 μL). Measurements were performed in triplicate.

With GeneSoC®, the output waveform was judged visually. If the fluorescent wavelength was determined to rise sharply above the baseline without falling back during measurement, a positive result was indicated, and if it remained at the baseline with no obvious rise, a negative result was indicated. For the LightCycler®, in addition to visual confirmation of a rise in the fluorescent signal, there is a built-in automated judgment function that provides the final result.

### Within-run repeatability

For within-run reproducibility testing, clinically isolated viral RNA at a concentration of $10^3$ copies/μL was used, and measurements were repeated for five consecutive runs. A comparison of Ct values was carried out using the second derivative maximum method to calculate the percent coefficient of variation (%CV).

### Between-run reproducibility

For the daily rate repeatability study, clinically isolated viral RNA at a concentration of $10^3$ copies/μL was used, and measurements were performed for three consecutive days. Comparison of Ct values was carried out using the second derivative maximum method to calculate the %CV.

### Reproducibility among different detection units

To test the reproducibility among different detection units, measurements were performed with three different concentrations of clinically isolated viral RNA ($10^4$, $10^3$, and 1 copy/μL) using three different detection units connected to the same GeneSoC® main unit. Ct values were compared using the second derivative maximum method to calculate the %CV.

**Table 3. GeneSoC® PCR conditions.**

| Condition | Temperature [˚C] | Time [s] |
|---|---|---|
| Reverse transcription | 50 | 90 |
| Hot start | 96 | 10 |
| Denaturation | 96 | 5 |
| Annealing | 60 | 8 |
| Cycle | | 50 |

**Table 4. LightCycler® 480 primer and probe arrangement.**

| Primer | Primer and Probe Arrangement (5′ to 3′) |
|---|---|
| Forward (10 μM) | AAATTTTGGGGACCAGGAAC |
| N2 Reverse (10 μM) | TGGCAGCTGTGTAGGTCAAC |
| Probe (5 μM) | FAM–ATGTCGCGCATTGGCATGGA–TAMRA |

## Method comparison study

For the method comparison study, residual samples from a total of 1,445 clinical specimens were used. Measurements were performed with the LightCycler® 480 and GeneSoC® systems, and result judgments were made using the applicable judgment method for each detection method (https://www.niid.go.jp/niid/ja/) [8].

## Statistical analysis

The positive and negative predictive rates were calculated for the comparison between methods in this study. The Microsoft Excel software (Microsoft Corporation, Redmond, WA, USA) was used.

# Results

## Analytical sensitivity

Results of the analytical sensitivity study using a dilution series of the standard showed that GeneSoC® detected the target sequence of E and RdRp genes at a concentration of 5 and 10 copies/μL, respectively (Fig 1A and 1B), whereas LightCycler® 480 detected the N2 gene at a concentration of 2 copies/μL (Table 6).

## Within-run repeatability

The %CV values acquired from five consecutive measurements of $10^3$ copies/μL RNA were 1.48% and 2.89% for the E and RdRp primers, respectively (Table 7). Both were within 3%, indicating a good reproducibility.

## Between-run reproducibility

The %CV values acquired from three consecutive days of measurements of $10^3$ copies/μL RNA were 1.84% and 0.84% for the E primer and RdRp primers, respectively (Table 8). Both were within 2%, indicating a good reproducibility.

**Table 5. LightCycler® 480 PCR conditions.**

| | Mode | Cycles | Temp. [˚C] | Time | Ramp Rate [˚C/s] | Acquisition Mode |
|---|---|---|---|---|---|---|
| Reverse transcription | None | 1 | 50 | 30 min | 4.4 | None |
| Denature | None | 1 | 95 | 15 min | 4.4 | None |
| PCR | Quantification | 45 | 95 | 15 s | 4.4 | None |
| | | 45 | 60 | 60 s | 2.2 | Single |
| Cooling | None | 1 | 40 | 30 s | 4.4 | None |

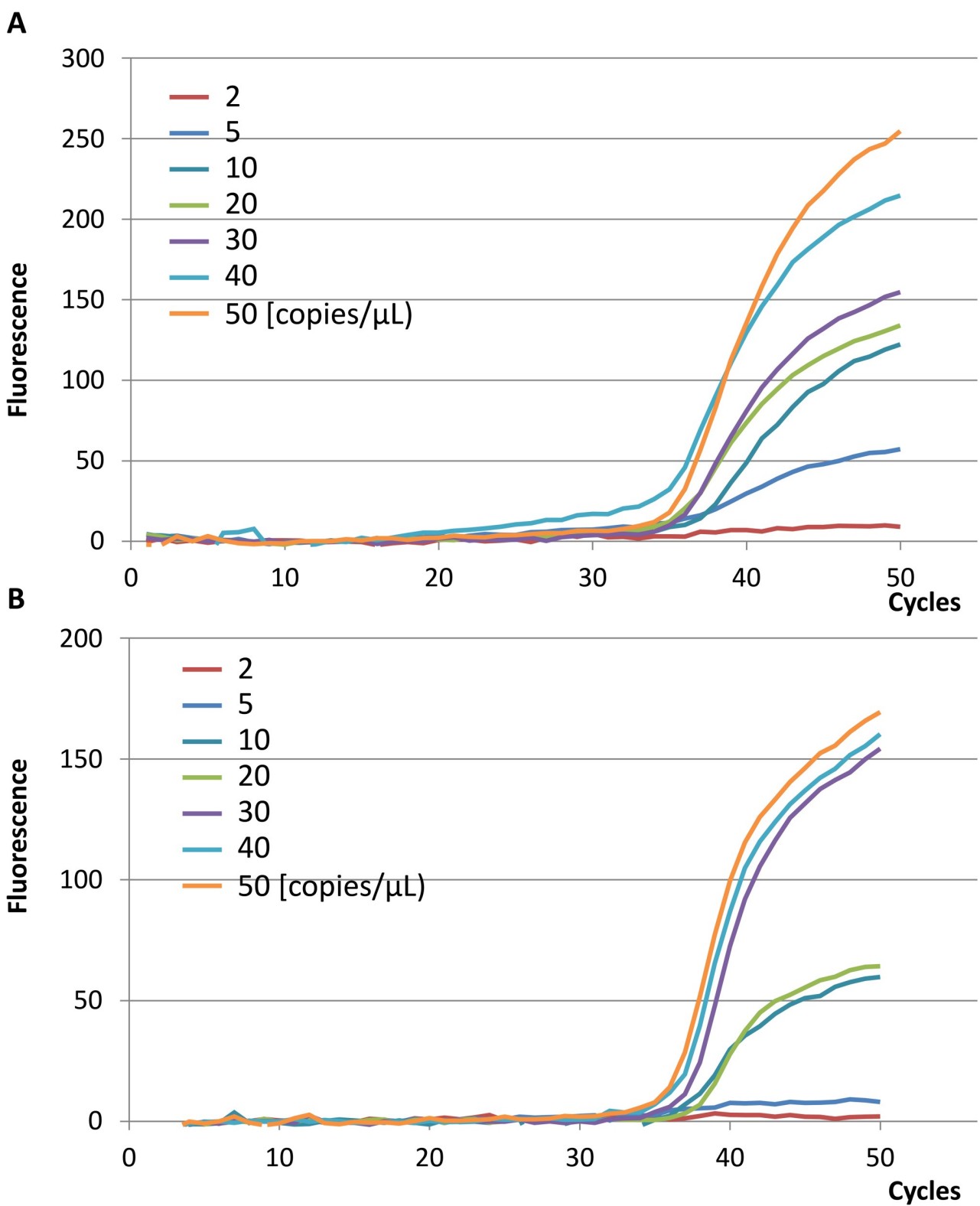

**Fig 1. Results of the measurement of SARS-CoV-2 RNA standards using the GeneSoC®.** E (A) and RdRp (B) Primers. The concentrations of the SARS-CoV-2 RNA standards were $5 \times 10^1$, $4 \times 10^1$, $3 \times 10^1$, $2 \times 10^1$, $1 \times 10^1$, $5 \times 10^0$, and $2 \times 10^0$ copies/μL. For $5 \times 10^1$, $4 \times 10^1$, $3 \times 10^1$, $2 \times 10^1$, and $1 \times 10^1$ copies/μL, a rising waveform was observed, and positive results were confirmed for both the E and RdRp primers. For $5 \times 10^0$ copies/μL, a rising waveform was only observed with the E primer. For $2 \times 10^0$ copies/μL, no rising waveform could be identified with both E and RdRp primers. E primer (E), RdRp primer (RdRp).

**Table 6. Comparison of the primer detection sensitivity of GeneSoC® and LightCycler®.**

| Primer (copies/µL) | $5 \times 10$ | $4 \times 10$ | $3 \times 10$ | $2 \times 10$ | $1 \times 10$ | $5 \times 10^0$ | $2 \times 10^0$ |
|---|---|---|---|---|---|---|---|
| N2 (LightCycler® 480) | + | + | + | + | + | + | + |
| E (GeneSoC®) | + | + | + | + | + | + | - |
| RdRp (GeneSoC®) | + | + | + | - | + | - | - |

### Reproducibility among different detection units

No major differences were observed between the detection units in the measurement of different concentrations of RNA using the E and RdRp primers (Table 9). The %CV values were as follows: (a) $10^4$ copies/µL, E primer 2.04% and RdRp primer 4.72%; (b) $10^3$ copies/µL, E primer 0.00% and RdRp primer 1.68%; and (c) $10^1$ copy/µL, the test results were negative for both the genes. For the detectable RNA concentrations ($10^4$ copies/µL and $10^3$ copies/µL), %CV was within 5% for both the genes, indicating no significant difference among detection units.

### Method comparison study

The method comparison of the GeneSoC® with the LightCycler® 480 system for testing 38 clinical samples revealed positive and negative agreement of 100% [(174/174) and (1271/1271), respectively] (Table 10).

## Discussion

This study evaluated the basic assay performance of the GeneSoC® system for the detection of SARS-CoV-2 RNA. The GeneSoC® system uses a microfluidic thermal cycling technology that can rapidly produce a nucleic acid amplification reaction as quickly as within 10–15 min. Precision study results were favorable for both within and between runs at RNA concentrations of $10^3$ copies/µL with a CV within 3%, and between units with a CV within 5%, indicating stable detection performance. The time required for SARS-CoV-2 RNA PCR testing can be reduced using the GeneSoC® system. When the GeneSoC® results of 1445 clinical samples were compared with those of LightCycler® 480, positive and negative agreement was 100%. This suggests that the GeneSoC® system could be effective as a rapid confirmatory diagnostic method for symptomatic cases normally having a high viral load.

**Table 7. Within-run reproducibility.**

| Ct values | | |
|---|---|---|
| n | E Primer | RdRp Primer |
| 1 | 30.0 | 32.0 |
| 2 | 31.0 | 34.5 |
| 3 | 30.0 | 34.0 |
| 4 | 30.0 | 34.0 |
| 5 | 30.0 | 34.0 |
| Average | 30.20 | 33.70 |
| SD | 0.45 | 0.97 |
| %CV | 1.48 | 2.89 |

Standard deviation (SD); Coefficient of variation (CV).

**Table 8. Between-run reproducibility.**

| Ct values | | |
|---|---|---|
| Day | E Primer | RdRp Primer |
| 1 | 32.00 | 34.00 |
| 2 | 31.00 | 34.00 |
| 3 | 31.00 | 34.50 |
| Average | 31.33 | 34.17 |
| SD | 0.58 | 0.29 |
| %CV | 1.84 | 0.84 |

Standard deviation (SD); Coefficient of variation (CV).

The GeneSoC® system uses a microfluidic thermal cycling technology to drive the nucleic acid amplification reaction. As a result, the amplification efficiency is almost the same, with analytical sensitivity not significantly lower as compared to standard PCR methods. The analytical sensitivity study revealed that the GeneSoC® system targeting E and RdRp genes was slightly inferior to the LightCycler® 480 system targeting the N2 gene, which is the method recommended by the National Institute of Infectious Diseases. Although the GeneSoC® system was shown to have a tendency toward false negatives when the viral load of a sample was very low, no clinical samples yielded a negative result. Virus multiplication and infectivity reportedly declines from the 9[th] day after the onset of illness [9], and the national guidance for the isolation of patients in Japan has been revised to reflect this. In a clinical setting, the need for negative confirmation testing is no longer mandatory. Accordingly, difficulty in negative confirmation with the GeneSoC® system is unlikely to be an impediment in using it in a clinical setting. Causes for clinical samples with a low viral load leading to a failure of viral detection include improper sample collection, specimen types, and collection timing in the clinical course. When COVID-19 is strongly suspected in pneumonia patients but the virus is not detected with nasopharyngeal swabs, the use of specimens from the lower respiratory tract, such as sputum samples, should be considered [10, 11].

For COVID-19, GeneSoC® is an effective method for the rapid detection of early-stage symptomatic cases with high viral load. The reproducibility among different detection units was confirmed to be favorable, and because multiple detection units can be fitted to a single main unit, the system can be used for the rapid and simultaneous measurement of multiple samples. Including the time required for nucleic acid extraction procedures, the test can be

**Table 9. Reproducibility of GeneSoC® among different detection units.**

| Ct values | | | | | | |
|---|---|---|---|---|---|---|
| Detection Unit | $10^4$ copies/μL | | $10^3$ copies/μL | | 1 copy/μL | |
| | E | RdRp | E | RdRp | E | RdRp |
| | Primer | Primer | Primer | Primer | Primer | Primer |
| 1 | 29.0 | 31.0 | 32.0 | 35.0 | - | - |
| 2 | 28.0 | 32.0 | 32.0 | 34.0 | - | - |
| 3 | 28.0 | 34.0 | 32.0 | 34.0 | - | - |
| Average | 28.33 | 32.33 | 32.00 | 34.33 | - | - |
| SD | 0.58 | 1.53 | 0.00 | 0.58 | - | - |
| %CV | 2.04 | 4.72 | 0.00 | 1.68 | - | - |

Standard deviation (SD); Coefficient of variation (CV).

**Table 10. Comparison of clinical sample detection between GeneSoC® and LightCycler® 480.**

| | | LightCycler® 480 | | |
| --- | --- | --- | --- | --- |
| | | Positive | Negative | Total |
| GeneSoC® | Positive | 174 | 0 | 174 |
| | Negative | 0 | 1274 | 1274 |
| | Total | 174 | 1274 | 1445 |

completed in as little as 40 min. Furthermore, a direct real-time RT-PCR method, based on exclusion of the interfering substances, are investigated. According to the preliminary study using a probe for the N2 gene, this method shows high sensitivity and the results are obtained in 20–30 min, including the time taken for the addition of reagents. Accordingly, the system would allow us to report test results in a rapid and real-time manner. Multiple specimen processing could reduce the burden on testing sites. When a simple and rapid procedure for RNA extraction, such as inhibitor inactivation, can be incorporated, the total assay time could be shortened to 20 min. Such an assay will become more useful, especially considering the recent trend of increasing numbers of COVID-19 cases as a resurgence of the epidemic has been observed, as well as the expected increase in the number of cases in the coming winter months, owing to seasonal factors in Japan.

For an emergency case, warranting immediate hospitalization and isolation, a chest CT scan is necessary regardless of test availability and its result. When rapidity is not an issue, the use of nucleic acid amplification detection methods with a higher analytical sensitivity, such as LightCycler® 480 targeting the N2 gene, should be considered.

In the GeneSoC® detection system, test results are judged based on a visual procedure analyzing the output waveform, which determines a final positive or negative result to be released. However, if the sample viral load is near the analytical sensitivity limit, judging the rising waveform can be difficult in some cases. In addition, currently, there are no positive or internal controls in use, and thus, a positive result due to contamination or a possible false negative result due to inhibitors cannot be ruled out. Based on these points, the development of clear judgment standards and automated judgment of results are desirable for the future. Currently, the development of a fully automated system of all processes, including automated extraction for the improvement of the analytic sensitivity of measurements and overall process speed, as well as automated judgment of the output waveform, is underway.

In conclusion, the GeneSoC® system is beneficial for a rapid, reliable, and highly sensitive real-time testing of SARS-CoV-2. Further applications in a wide range of situations in medical settings are warranted, along with improvements in the performance of the assay.

## Acknowledgments

The researchers received extensive assistance with the implementation of the GeneSoC® system from Kyorin Pharmaceutical Co., Ltd.

## Author Contributions

**Conceptualization:** Satomi Asai.

**Formal analysis:** Ryosuke Watanabe, Atsuko Masukawa, Kazuo Umezawa.

**Funding acquisition:** Satomi Asai.

**Investigation:** Ryosuke Watanabe, Atsuko Masukawa, Miki Miyazawa, Kazumi Ohtagawa, Mend-Amar Ravzanaaadii.

**Project administration:** Satomi Asai.

**Supervision:** Kazuo Umezawa, Hayato Miyachi.

**Validation:** Mika Doi, Haruyo Atsumi.

**Visualization:** Hidehumi Kakizoe, Hirofumi Saeki.

**Writing – review & editing:** Satomi Asai.

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
