## [Decision Letter · Decision Letter 0]

30 Dec 2020

PONE-D-20-38418

Evaluation of the basic assay performance of the GeneSoC® Rapid PCR Testing System for detection of Severe Acute Respiratory Syndrome Coronavirus 2

PLOS ONE

Dear Dr. Asai,

Thank you for submitting your manuscript to PLOS ONE. After careful consideration, we feel that it has merit but does not fully meet PLOS ONE’s publication criteria as it currently stands. Therefore, we invite you to submit a revised version of the manuscript that addresses the points raised during the review process.

The comments raised by the reviewers seem minor. I hope you can revise your manuscript easily.

We look forward to receiving your revised manuscript.

Kind regards,

Etsuro Ito

Academic Editor

PLOS ONE

Journal Requirements:

Reviewers' comments:

Reviewer's Responses to Questions

**Comments to the Author**

1. Is the manuscript technically sound, and do the data support the conclusions?

Reviewer #1: Yes

Reviewer #2: Yes

2. Has the statistical analysis been performed appropriately and rigorously? 

Reviewer #1: Yes

Reviewer #2: Yes

3. Have the authors made all data underlying the findings in their manuscript fully available?

Reviewer #1: Yes

Reviewer #2: Yes

4. Is the manuscript presented in an intelligible fashion and written in standard English?

Reviewer #1: Yes

Reviewer #2: Yes

5. Review Comments to the Author

Reviewer #1: This paper deals with the evaluation of a new PCR system, GeneSoc, for detection of SARS-CoV-2 from clinical specimens. The authors present a careful and thorough analysis of this system while comparing with another system, LightCycler 480, that is the major testing tool used in Japan. The aim addressed is interesting and important.

I would recommend it for acceptance after the minor points listed below.

Comments:

1. GeneSoc PCR system uses the primers and probes targeted to two genes (E and RdRp genes). Why these genes were chosen? Does the employment of these genes in this system have an advantage over LightCycler 480 system where one gene (N2 gene) is used as primer and probe?

2. As shown in Table 6, LightCycler 480 system was superior to GeneSoc system in term of the sensitivity for the detection of SARS-CoV-2. It is important not to miss the patients with a low viral load for preventing the spread of SARS-CoV-2 infection. Is there any room for improvement in the sensitivity of viral detection?

3. In specimen processing, the authors used a QIAamp Viral RNA Mini Kit. How long does it take to extract viral RNA? Is there a possibility of further time saving?

Reviewer #2: In this study, RT-PCR assay for SARS-CoV-2 by GeneSoC® was completed in 15 min using the system described by the authors. The positive conformity ratio for SARS-CoV-2 of the GeneSoC® was similar to that obtained with the LightCycler®480 system. The quality control guidelines of this system were well described. The system is appropriate for medical and clinical use.

The following minor changes are necessary so that this thesis is accepted.

Q1. You targeted only the N2 region on the LightCycler® 480, why did you not target the N1 region.

Q2. RNA extraction takes more than 30 min. further, development of such as the direct RT-PCR assay method is needed. Do you have any other suggestions?

Q3. How was the reproducibility of the three negative samples?

Q4. Please emphasize that it is more useful even if it is less sensitive than the light cycler.

6. PLOS authors have the option to publish the peer review history of their article (what does this mean?). If published, this will include your full peer review and any attached files.

Reviewer #1: No

Reviewer #2: No

---

## [Author Response · Author response to Decision Letter 0]

21 Feb 2021

Responses to the reviewer’s comments

This paper deals with the evaluation of a new PCR system, GeneSoc, for detection of SARS-CoV-2 from clinical specimens. The authors present a careful and thorough analysis of this system while comparing with another system, LightCycler 480, that is the major testing tool used in Japan. The aim addressed is interesting and important.

I would recommend it for acceptance after the minor points listed below.

Comments:

1. GeneSoc PCR system uses the primers and probes targeted to two genes (E and RdRp genes). Why these genes were chosen? Does the employment of these genes in this system have an advantage over LightCycler 480 system where one gene (N2 gene) is used as primer and probe?

Response

As mentioned in the main text, the N region is recommended by the (Japanese) National Institute of Infectious Diseases. Two types of primers and probes, N and N2, are available for this region. The system presents a positive result if either N or N2 is positive. However, owing to the better sensitivity obtained with N2, many facilities in Japan only use the N2 probe for detection. We have explained this in the revised manuscript for the benefit of the readers (Lines 129-130). Although tests using the N gene are popular in Japan, when GeneSoC® was first developed in 2020, tests using E and RdRp genes were also common in other countries, and Roche has also released PCR reagents for these genes in the Japanese market. Meanwhile, GeneSoC® has also developed an RT-PCR for the N2 region, which is being prepared for launch. This test has been demonstrated to have a better sensitivity and specificity, which has been included at the end of the Discussion section (Lines 259-266).　Of note, the primers and probes are designed, developed, and used by each manufacturer and institution; importantly, the use of primers and probes for E and RdRp genes has an equivalent advantage over the use of the primer and probe for a single (N2) gene in the LightCycler 480 system.

2. As shown in Table 6, LightCycler 480 system was superior to GeneSoc system in term of the sensitivity for the detection of SARS-CoV-2. It is important not to miss the patients with a low viral load for preventing the spread of SARS-CoV-2 infection. Is there any room for improvement in the sensitivity of viral detection? 

Response

As you pointed out, when we were preparing this manuscript, this system was not as sensitive as the LightCycler® 480. Because the kit has since been optimized by the manufacturer, all positive samples were retested. We also confirmed that the sensitivity for the Standards has increased to detect 2 copies/assay. The results showed that the E and RdRp probes detected 5 and 10 copies/assay, respectively. The LightCycler® 480 detected 2 copies/assay. Based on these results, we have replaced the Table, the results including the graphs, and the discussion of the results (Lines　33, 38, 140-141, 188-191, 195-197, 199, 237, 253-257, and 259-266, Table 6, Table 10, and Figure 1). Importantly, the difference in the sensitivity was found not to have any implications in the context of clinical applications (Lines 194-201).

　　

　

3. In specimen processing, the authors used a QIAamp Viral RNA Mini Kit. How long does it take to extract viral RNA? Is there a possibility of further time saving? 

Response

RNA extraction using a column-based method, such as the QIAamp Viral RNA Mini Kit, requires 35–50 minutes. RT-PCR of the extracted RNA can be completed in less than 15 minutes with GeneSoC®. Thus, the process from extraction to the acquisition of results takes approximately 60 minutes. Thus, GeneSoC® may be more useful than the LightCycler® 480 system in situations where rapid testing is required, particularly with patients who have been brought in to the emergency room or are suspected of being infected. In fact, we use the GeneSoC® system in such cases at our facility. A direct real-time RT-PCR method, based on exclusion of the interfering substances, using a probe for the N2 gene region has been developed for GeneSoC®. Using this method, the sensitivity is the same as that of the current method (high sensitivity), and the results can be obtained in 25–30 minutes, including the time taken for the addition of reagents (Lines 281-284). This method is scheduled to be launched soon. After incubating clinical specimens at room temperature for 10 minutes, the GeneSoC® RT-PCR method can be completed in approximately 15 minutes, and all the processes will be completed in 25 minutes (Data and comments are not shown in this manuscript and would be presented in our next paper, which is under preparation).

The following minor changes are necessary so that this thesis is accepted.

Q1. You targeted only the N2 region on the LightCycler® 480, why did you not target the N1 region.

Response

 Please refer to our response to Comment 1.

As mentioned in the main text, the N region is recommended by the (Japanese) National Institute of Infectious Diseases. Two types of primers, N and N2, are available for this region. The system presents a positive result if either N or N2 is positive. However, owing to the better sensitivity obtained with N2, many facilities in Japan only use the N2 primer for diagnosis. We have explained this in the revised manuscript for the benefit of the readers (Lines 129-130).

Q2. RNA extraction takes more than 30 min. further, development of such as the direct RT-PCR assay method is needed. Do you have any other suggestions?

Response. 

Please refer to our response to Comment 3.

A direct real-time RT-PCR method using a probe for the N2 gene region has been developed for GeneSoC®. According to this method, the sensitivity is the same as that of the current method (high sensitivity), and the results can be obtained in 25–30 minutes, which includes the time taken for the addition of reagents (Lines 281-284). This method is scheduled to be launched soon. After incubating clinical specimens at room temperature for 10 minutes, the GeneSoC® RT-PCR method can be completed in approximately 15 minutes, and all the processes will be completed in 25 minutes (Data and comments are not shown in this manuscript and would be presented in our next paper, which is under preparation).

Q3. How was the reproducibility of the three negative samples?

Response

No samples tested negative in the retesting using the improved kit that is available now.

Q4. Please emphasize that it is more useful even if it is less sensitive than the light cycler.

Response　

Thank you very much for this suggestion. In our discussion of the usefulness of GeneSoC®, we have emphasized that it offers high speed, reliability, and sensitivity (Line 309-310).

---

## [Editor Report · Decision Letter 1]

26 Feb 2021

Evaluation of the basic assay performance of the GeneSoc® Rapid PCR Testing System for detection of Severe Acute Respiratory Syndrome Coronavirus 2

PONE-D-20-38418R1

Dear Dr. Asai,

We’re pleased to inform you that your manuscript has been judged scientifically suitable for publication and will be formally accepted for publication once it meets all outstanding technical requirements.

Kind regards,

Etsuro Ito

Academic Editor

PLOS ONE

---

## [Editor Report · Acceptance letter]

22 Mar 2021

PONE-D-20-38418R1 

Evaluation of the basic assay performance of the GeneSoc^®^ Rapid PCR Testing System for detection of Severe Acute Respiratory Syndrome Coronavirus 2 

Dear Dr. Asai:

I'm pleased to inform you that your manuscript has been deemed suitable for publication in PLOS ONE. Congratulations! Your manuscript is now with our production department. 

Kind regards, 

on behalf of

Prof. Etsuro Ito 

Academic Editor

PLOS ONE